# Indirect Genetic Effects of ADIPOQ Variants on Lipid Levels in a Sibling Study of a Rural Chinese Population

**DOI:** 10.3390/genes13010161

**Published:** 2022-01-17

**Authors:** Zechen Zhou, Yujia Ma, Xiaoyi Li, Zeyu Yan, Kexin Ding, Han Xiao, Yiqun Wu, Dafang Chen, Tao Wu

**Affiliations:** Department of Epidemiology and Biostatistics, School of Public Health, Peking University Health Science Center, Beijing 100191, China; 2011110141@bjmu.edu.cn (Z.Z.); 13260033986@163.com (Y.M.); lvxi520@pku.edu.cn (X.L.); yanzysy@163.com (Z.Y.); 15801298733@163.com (K.D.); gesangmeiduo@pku.edu.cn (H.X.); qywu118@163.com (Y.W.); twu@bjmu.edu.cn (T.W.)

**Keywords:** ADIPOQ, indirect genetic effect, lipid level, sibling study

## Abstract

Variations in lipid levels are the result of combinations of genetic and environmental factors. We aim to investigate the indirect effect between siblings of the three polymorphisms of ADIPOQ on serum lipid levels in rural Chinese populations. A total of 2571 sibling pairs were enrolled as study participants. A generalized estimating equation was used to accommodate a family-based design. We used stratified analysis to detect sex combination differences in the indirect genetic effect. We found a significant association between the number of altered risk alleles of rs182052 and ego lipid levels of TG (β = 0.177, *P* = 0.003), TC (β = 0.140, *P* = 0.004) and LDL-C (β = 0.098, *P* = 0.014). Ego and altered genotypes of rs182052 demonstrated a joint effect on ego lipid levels of TC (β = 0.212, *P* = 0.019), HDL-C (β = 0.099, *P* = 0.002) and LDL-C (β = 0.177, *P* = 0.013) in recessive inheritance mode. In opposite-sex siblings, the altered GG genotype of rs182052 increased the ego lipid levels. Thus, our findings demonstrate that ADIPOQ has an indirect genetic effect on lipid levels in sibling pairs, and there are sex-combination differences in the indirect genetic effect in siblings.

## 1. Introduction

In the last decade, the prevalence of dyslipidemia has increased significantly [1], while the awareness rate, treatment rate and control rate of dyslipidemia in adults has been low [2]. Dyslipidemia is an important component of metabolic syndrome and an important risk factor for atherosclerotic cardiovascular disease. Studying the influencing factors of blood lipid levels is conducive to predicting the occurrence of dyslipidemia and its early intervention, which helps the prevention and treatment of cardiovascular metabolic diseases. Similar to most chronic noncommunicable diseases, variations in lipid levels are the result of combinations of genetic and environmental factors. Lipid traits, mainly triglycerides (TGs), total cholesterol (TC), high-density lipoprotein cholesterol (HDL-C) and low-density lipoprotein cholesterol (LDL-C), are highly heritable, with estimated heritability ranging from 20% to 70% [3]. Genome-wide association studies (GWASs) have identified more than 100 SNPs associated with lipid traits, many of which are shared across more than one lipid trait [4,5]. However, the mechanisms through which these single nucleotide polymorphisms (SNPs) act on each trait are poorly understood. In terms of environmental factors, many lifestyles can affect lipid levels, including physical inactivity, diets with high positive energy-intake balance due to high-fat or high-glycemic index and alcohol consumption [6,7].

Genetic and environmental factors are not independent. On the one hand, there are complex interactions between them. On the other hand, many lifestyle risk factors also have a genetic basis. For example, GWASs have identified a mass of loci related to diet [8], cigarette smoking [9], alcohol consumption [10], exercise and sleep [11]. Moreover, many of an individual’s environmental risk factors are modifiable and susceptible to the influence of others. In 2018, Augustine Kong et al. found that parental alleles not transmitted to offspring could also have significant effects on offspring traits and explained the phenomenon as a genetic nurture effect [12]. The genetic nurture effect refers to the indirect effect of parental genotype on offspring traits by shaping the growing environment or lifestyle of the offspring. In addition to the parent–child relationship, indirect genetic effects can also occur between other people with close relationships and shared living environments, such as siblings, spouses and close friends. Indirect genetic effects provide an entirely new way of understanding how closely genetic and environmental factors interact with each other.

Previous studies have indicated that lipid levels are significantly correlated among sibling pairs. Lee A Pyles estimated that the correlation between siblings in LDL-C is 0.375, HDL-C is 0.34 and TG is 0.22 [13]. This correlation could be due to sibling pairs having similar genes or sharing the same environment or lifestyle. By analyzing the genotype of siblings, we can decompose the genetic effect into direct effect and indirect effect and estimate the extent to which a person’s genotype can influence their siblings’ lipid levels through their common environmental phenotypes. Similar to other association studies, indirect genetic effects can be carried out by genome-wide approaches, such as polygenetic scores and candidate gene strategies. For genes known to be associated with disease phenotypes, the analysis of their indirect genetic effects is conducted to understand the function and mechanism of the targeted genes.

Several studies have found that multiple SNPs in the adiponectin gene (ADIPOQ) might be related to lipid levels and dyslipidemia [14]. ADIPOQ encodes adiponectin, which has a modular structure consisting of a collagen-like N-terminal domain and a C-terminal globular domain and is a specific protein secreted by adipose tissue. Genetic variants in the gene encoding adiponectin have been reported to be associated with adiponectin levels in several genome-wide linkage and association studies [15,16]. It has been reported that ADIPOQ can directly affect lipid levels through physiological pathways, including enhanced glucose and fatty acid uptake [17], as well as oxidation in skeletal muscle [18]. On the other hand, ADIPOQ may also indirectly affect blood lipid levels by influencing individual eating behavior through central effects in the brain [19] or relaxant effects on smooth muscle [20]. Cross-sectional findings have demonstrated that siblings are moderately similar in their eating behavior [21]. Therefore, we hypothesize that the variants of the genotype of ADIPOQ have indirect effects on sibling lipid levels.

In the current study, we aimed to analyze the correlation of serum lipid levels in sibling pairs and to investigate whether variants of ADIPOQ have an indirect effect between siblings in rural Chinese populations.

## 2. Materials and Methods

### 2.1. Study Design and Participants

The study participants were included from the Fangshan Family-based Ischemic Stroke Study in China (FISSIC) [22]. FISSIC is an ongoing community-based case-control genetic epidemiological study that started in June 2005, which enrolls families in Fangshan District, a rural area located southwest of Beijing, China. A total of 7718 samples distributed across 3398 families were recruited for the study using the proband-initiated contact method. The inclusion criteria for sibling pairs were as follows: (1) age older than 18 years old at enrollment; (2) variables of sex, age or TG, TC, HDL-C or LDL-C were not missing in sibling pairs; (3) genotyping rate ≥90% in both of the siblings; and (4) subjects without single-gene hereditary disease or cancer. As a result, we recruited 2571 sibling pairs as our study participants.

This study was approved by the Ethics Committee of the Peking University Health Science Center (Approval number: IRB00001052-13027), and written informed consent was provided by all participants.

### 2.2. Assessment of Lipid Levels and Covariates

Laboratory tests of serum lipid levels, including TG, TC, HDL-C and LDL-C, were performed by qualified technicians from the Laboratory of Molecular Epidemiology in the Department of Epidemiology at Peking University.

Demographic information, lifestyle risk factors and self-reported medical history were collected through a face-to-face questionnaire survey by trained staff.

Physical measurements, including height, weight, waist circumference (WC) and blood pressure, were performed by trained investigators. Body mass index (BMI) was calculated as weight (kg) divided by squared height (m^2^). Fasting blood glucose (FBG) and hemoglobin A1c were also collected. Obesity was defined as BMI ≥ 28 kg/m^2^. Hypertension was defined as a self-reported history, systolic blood pressure ≥ 140 mm Hg, diastolic blood pressure ≥ 90 mm Hg and/or use of antihypertensive medications. Type 2 diabetes (T2D) was defined according to self-reported records or abnormal glycemic markers (fasting blood glucose ≥ 7.0 mmol/L or hemoglobin A1c ≥ 6.5%). Dyslipidemia was defined according to self-reported records or abnormal glycemic markers (TG ≥ 2.27 mmol/L, TC ≥ 6.45 mmol/L, HDL-C < 1.04 mmol/L or LDL-C ≥ 4.9 mmol/L).

### 2.3. Genotyping

DNA was extracted using a LabTurbo 496-Standard System (TAIGEN Bioscience Corporation, Taiwan, China). In addition, the purity and concentration of DNA were measured using ultraviolet spectrophotometry. Furthermore, the genomic DNA sample was genotyped with the time of flight mass spectrum using the MassARRAY^®^ System (Agena Bioscience, San Diego, CA, USA). We used two negative (blanks) and three positive controls to control the quality of the genotyping process, and the results were satisfied. We also chose 5% samples randomly for repeat analysis to verify the reproducibility of the genotyping data.

Three SNPs associated with lipid metabolism of ADIPOQ were selected, including rs16861194, rs266729 and rs182052. To control the quality of the genotype, the call rate was examined (>95%) to check the accuracy of genotyping. The minimum allele frequency (MAF) of SNPs was examined to be ≥5%. The SNP Hardy–Weinberg test (*P* ≥ 0.05 after Bonferroni correction) was performed in randomly selected individuals, one from each sibling pair. As a result, all three SNPs passed the quality control procedure. The specific information of the three SNPs is shown in Appendix A.

### 2.4. Statistical Analysis

Categorical variables, such as sex, ethnicity, hypertension, type 2 diabetes, dyslipidemia, smoking, drinking, moderate-high intensity physical activity and genotypes, were described as percentages in the case and control groups. Continuous variables, including age, BMI, WC, SBP, DBP, FBG, TGs, TC, HDL-C and LDL-C, are presented as the mean ± standard deviation (SD).

According to previous literature [23], we refer to the individual whose behavior may be affected as the ego and the individual who may be affecting them as the alter to better distinguish between the two in a sibling pair. In other words, the blood lipid level of ego is the outcome of our study, while the genotype of ego and alter is the influencing factor of the outcome.

Siblings share on average 50% of their genes identical by descent, so the association between altered genes and ego traits is highly confounded by the ego gene. We wanted to explore whether altered genotypes can affect ego traits under the control ego genotypes. Therefore, for the establishment of the model, we regressed the ego’s lipid levels (Y) on the ego’s own genotype, the altered genotype and covariates, including sex, age and ethnicity:Y = β_0_ + β_1_Genotype_ego_ + β_2_Genotype_alter_ + β_3_Covariates + ε

In the above formula, the parameter we are most interested in is β_2_, which represents the indirect effect of the genotype on lipid levels. Parameter ε refers to the random error. To model the association between the genotypes and traits, a generalized estimating equation (GEE) was used to accommodate a family-based design. To control for confounders, we adjusted for sex, age and ethnicity as covariates. The β value (β) of the regression model was derived to represent the genetic effect for an additional risk allele.

We first employed an additive genetic model to investigate the association of risk alleles with the clinical traits of siblings. For the SNPs and traits found to have significant associations, we further explored the joint effect of ego genotypes and altered genotypes on ego traits and assessed the possible mode of inheritance of the SNPs. In addition, we stratified the analysis by the sex combination of sibling pairs (same-sex siblings, opposite-sex siblings). We also conducted stratified analyses by the specific sex of siblings (male-male, male-female or female-female).

All statistical analyses were conducted using R version 4.0.3 (R Foundation for Statistical Computing, Vienna, Austria). P values of less than 0.05 (two-sided) were considered statistically significant.

## 3. Results

### 3.1. Description of the Study Population

The distribution of characteristics for the first-born and second-born siblings is presented in Table 1. Compared with the first-born siblings, the second siblings had a higher BMI (*P* = 0.033) and DBP (*P* = 0.001), a higher prevalence of drinking (*P* = 0.004), lower SBP (*P* < 0.001) and FBG (*P* = 0.010), and a lower prevalence of hypertension and T2D. The second-born siblings had similar mean levels as their siblings at TG, TC HDL-C and LDL-C and had a similar genotypic distribution of ADIPOQ polymorphisms. In addition, we carried out a correlation analysis of clinical traits between siblings, and the results are shown in Appendix A. Siblings demonstrated high intraclass correlation coefficients of 0.620 for TC, 0.670 for HDL-C, and 0.533 for LDL-C but demonstrated low intraclass correlation coefficients of other traits.

### 3.2. The Association between ADIPOQ Polymorphisms and Clinical Traits in Siblings

After adjusting for the known covariates (sex, age and ethnicity), both ego alleles and altered alleles of rs182052 were found to be significantly associated with TG (ego alleles: β = 0.117, *P* = 0.024; alter alleles: β = 0.177, *P* = 0.003), TC (ego alleles: β = 0.092, *P* = 0.023; alter alleles: β = 0.140, *P* = 0.004) and LDL-C (ego alleles: β = 0.064, *P* = 0.050; alter alleles: β = 0.098, *P* = 0.014) (See Table 2). We found that alter alleles of rs182052 were significantly associated with HDL-C (β = 0.040, *P* = 0.019), alter alleles of rs266729 were significantly associated with WC (β = 1.583, *P* = 0.027) and DBP (β = 1.713, *P* = 0.020), while the ego alleles showed no significant association.

### 3.3. Joint Effect of Ego and Altered Genotypes of rs182052 on Lipid Levels

Based on the above results, we further explored the association between rs182052 and lipid levels in siblings. The joint effect of ego and the altered genotype of rs182052 on lipid levels based on the nine genotype combination groups is shown in Table 3. Compared with both the AA genotype of rs182052 in ego and alter, those who had both the GG genotype of rs182052 in ego and alter showed significantly higher levels in TC (β = 0.212, *P* = 0.019), HDL-C (β = 0.099, *P* = 0.002) and LDL-C (β = 0.177, *P* = 0.013) after adjusting for the known covariates.

### 3.4. The Association between rs182052 and Lipid Levels in Siblings Stratified by Sex

The above results suggest that the mutation of rs182052 may be recessive in the mode of inheritance. Based on the recessive mode, we investigated associations between the altered genotype of rs182052 and lipid levels stratified by sex combination of sibling pairs after adjustment for ego genotype and all covariates, and the results are shown in Table 4. We observed a significant association of the altered GG genotype of rs182052 with the levels of HDL-C (β = 0.067, *P* = 0.005) and LDL-C (β = 0.131, *P* = 0.024) in siblings of opposite sexes. We did not find any significant genetic association in the same-sex sibling group.

Based on the significant associations found in opposite-sex siblings, we further stratified the opposite-sex siblings into two specific sex combination types (male-ego with female-alter, female-ego with male-alter), and the result of association analysis is shown in Table 5. In the male ego with female alteration group, we observed a significant association of the altered GG genotype of rs182052 with the levels of TG (β = −0.230, *P* = 0.033) and HDL-C (β = 0.091, *P* = 0.021).

## 4. Discussion

In this study, we found that there are significant associations between the count of the alter risk allele of rs182052 and some lipid levels, while the coefficients are small. Ego and alter genotypes of rs182052 show a joint effect on the lipid levels in the recessive inheritance mode. In opposite-sex siblings, the alter GG genotype of rs182052 increases the level of HDL-C and LDL-C. The GG genotype of rs182052 of sisters can significantly increase the level of HDL-C and decrease the level of TG in males.

These findings suggest that the variants rs182052 of ADIPOQ have indirect genetic effects on the sibling’s lipid levels. Interestingly, the size of the indirect effect is even more than the direct effect (TG: 0.177 vs. 0.117, TC: 0.140 vs. 0.092, HDL-C: 0.040 vs. 0.009, and LDL-C: 0.098 vs. 0.064). The results suggest that the ADIPOQ gene may have a greater effect on lipid levels through influencing the environment and behavior habits than through direct physiological pathways. Although there is little research investigating indirect genetic effects of candidate genes on lipid levels between siblings, our findings are consistent with some related studies. For the indirect genetic effect on lipid levels, Kong et al. found that the parental polygenic score of educational attainment has a moderate genetic nurture on the childrens’ HDL level [12]. Using the structural equation modeling (SEM) approach, Anne E. Justice et al. identified five SNPs that have an indirect pathway effect on triglycerides through nearby methylation sites [24]. The results suggest that indirect genetic effects may play a role by influencing omics markers. Joint effects of ego and alter genotype on ego lipid levels have not been described in prior studies. Our results of the joint effect would considerably improve the identification of people with elevated lipid levels. Considering sibling pairs as a whole to assess one’s genetic risk of dyslipidemia may be an attractive approach for primary prevention.

We found sex-combination differences in siblings of the indirect genetic effect of rs182052. Opposite-sex siblings demonstrated significant associations between alter genotypes and ego lipid levels, while same-sex siblings did not. This result is inconsistent with some previous studies. Sara Pereira found that same-sex siblings aged 9–20 years demonstrated higher resemblance in HDL-C and other metabolic syndrome (MS) markers than opposite-sex siblings [25]. The inconsistency may result from differences in population characteristics, such as age distribution. No significant differences in lipid levels between same-sex siblings and opposite-sex siblings were found in our study, as is shown in Appendix A. We can further find clues in the research on behavior and social relationship characteristics of the elderly. Hiroko Akiyama indicated that older men tend to have closer contact with and receive more support from their sisters than their brothers [26]. Therefore, older men are easier to be influence by their female siblings on their behaviors and environment, which can explain our finding that the female alter GG genotype of rs182052 is significantly associated with male ego levels of TG and HDL-C. 

Our findings of the indirect genetic effect of rs182052 suggests that the polymorphism of rs182052 may influence individual behaviors, which may further influence their sibling’s behaviors or environment. The rs182052 polymorphism is located in intron 1 of the ADIPOQ promoter region, with the minimum gene frequencies (MAFs) greater than 5% in most populations of the 1000 Genome Project. Variants of rs182052 were found to be associated with different circulating adiponectin levels [27], and a study carried out in a Chinese population reported that rs182052 was related with low expression levels of adiponectin [28]. The mechanisms by which adiponectin levels regulate food intake remain unclear, but a number of epidemiological studies have investigated the association between levels of adiponectin and eating behavior. Human adiponectin levels were observed to be altered in eating disorders, such as anorexia nervosa, binge eating disorder and bulimia nervosa [29,30]. Kerstin Rohde found that genetic variants of the ADIPOQ locus has a potential influence on eating behavior, including disinhibition and hunger [31]. In addition, there are some reports of adiponectin association with mental disorders, such as anxiety disorder [32], trauma-related and stressor-related disorders [33]. The mental disorder can also affect eating behavior. Our results may provide insights into the function of adiponectin as a potential regulator of food intake. 

This study has some limitations. In our hypothesis, indirect genetic effect occurs between people who are closely related. Therefore, the effect size is supposed to be affected by the degree of intimacy between siblings. For example, the frequency of the contact between them, whether they live together in the same house and what they think of each other. There is no such variable in our database, and this may affect our estimation of indirect genetic effects. Nevertheless, our study samples were from Fangshan, a rural area with big family aggregation and weak population mobility, thus the sibling pairs collected were all in a stable close relationship, which could limit the confounding factors. On the other hand, the study samples were from a restricted area in northern China, and the indirect genetic effect of siblings may be influenced by local geography and cultural characteristics, thus the extrapolation of the conclusions of this study is limited. Finally, to elucidate the mechanism of indirect genetic effects in siblings, we need to further analyze environmental variables between siblings to investigate whether variants of ADIPOQ have a mediation effect on the shared environment of siblings.

## 5. Conclusions

In conclusion, we observed that genotypes of ADIPOQ were associated to blood lipid levels of siblings, and there were sex-combination differences in the association. Our findings suggest that ADIPOQ has an indirect genetic effect on lipid levels in sibling pairs. The results provide new evidence of factors affecting lipid levels, which can facilitate the understanding of the genetic basis of numerous risk factors of cardiovascular metabolic disease. In addition, our study provides guidance towards future studies of the indirect effects between sibling pairs.

## Figures and Tables

**Table 1 genes-13-00161-t001:** Demographic and clinical variables of the subjects of study.

Variables	Total (Mean ± SD or (%))	First-Born Sib (Mean ± SD or (%))	Second-Born Sib (Mean ± SD or (%))	*P*
N	3612	1806	1806	
Sex				
Female	1780 (49.3)	917 (50.8)	863 (47.8)	0.078
Male	1832 (50.7)	889 (49.2)	943 (52.2)	
Age, years	58.16 ± 8.21	61.62 ± 7.50	54.70 ± 7.39	<0.001
Ethnic				
Han	3530 (97.7)	1764 (97.7)	1766 (97.8)	0.560
Others	57 (1.6)	27 (1.5)	30 (1.7)	
Unknown	25 (0.7)	15 (0.8)	10 (0.6)	
BMI, kg/m^2^	26.24 ± 4.30	26.08 ± 4.78	26.39 ± 3.75	0.033
WC, cm	91.91 ± 9.79	92.02 ± 9.90	91.81 ± 9.68	0.538
SBP, mmHg	138.88 ± 20.11	141.34 ± 19.98	136.42 ± 19.95	<0.001
DBP, mmHg	82.80 ± 11.75	82.12 ± 11.72	83.47 ± 11.75	0.001
FBG, mmol/L	5.85 ± 2.75	5.97 ± 2.88	5.73 ± 2.60	0.010
TG, mmol/L	1.50 ± 1.22	1.50 ± 1.25	1.50 ± 1.19	0.948
TC, mmol/L	3.09 ± 1.11	3.10 ± 1.13	3.07 ± 1.10	0.549
HDL-C, mmol/L	0.92 ± 0.39	0.93 ± 0.40	0.92 ± 0.38	0.697
LDL-C, mmol/L	2.15 ± 0.88	2.15 ± 0.89	2.15 ± 0.88	0.917
Obesity	1033 (28.7)	498 (27.7)	535 (29.7)	0.192
Hypertension	2645 (73.3)	1436 (79.6)	1209 (67.1)	<0.001
T2D	1675 (46.4)	920 (50.9)	755 (41.8)	<0.001
Dyslipidemia	2706 (74.9)	1355 (75.0)	1351 (74.8)	0.908
Smoking	1695 (47.3)	845 (47.3)	850 (47.4)	0.944
Drinking	1357 (38.0)	635 (35.6)	722 (40.4)	0.004
Excercise	2149 (59.5)	1072 (59.4)	1077 (59.6)	0.892
rs16861194				
AA	2498 (69.2)	1245 (68.9)	1253 (69.4)	0.944
AG	983 (27.2)	496 (27.5)	487 (27.0)	
GG	131 (3.6)	65 (3.6)	66 (3.7)	
rs266729				
CC	1843 (51.0)	915 (50.7)	928 (51.4)	0.637
CG	1490 (41.3)	744 (41.2)	746 (41.3)	
GG	279 (7.7)	147 (8.1)	132 (7.3)	
rs182052				
AA	1005 (27.8)	515 (28.5)	490 (27.1)	0.619
AG	1729 (47.9)	860 (47.6)	869 (48.1)	
GG	878 (24.3)	431 (23.9)	447 (24.8)	

SD = standard deviation, N = number of individuals.

**Table 2 genes-13-00161-t002:** The associations with clinical traits for the ego and alter alleles.

Trait	SNPs	Ego Alleles		Alter Alleles	
		β(95%CI)	*P*	β(95%CI)	*P*
BMI	rs16861194	0.199(−0.152–0.551)	0.267	−0.276(−0.916–0.363)	0.397
	rs266729	0.054(−0.223–0.330)	0.703	0.367(−0.136–0.869)	0.153
	rs182052	0.376(0.074–0.678)	0.015	0.162(−0.166–0.491)	0.333
WC	rs16861194	−0.062(−0.812–0.687)	0.870	−1.453(−3.348–0.441)	0.133
	rs266729	0.312(−0.403–1.026)	0.393	1.583(0.176–2.990)	0.027
	rs182052	0.854(0.046–1.662)	0.038	0.654(−0.310–1.619)	0.184
SBP	rs16861194	1.023(−0.599–2.645)	0.217	−0.419(−3.939–3.100)	0.815
	rs266729	0.776(−0.619–2.170)	0.276	2.562(−0.181–5.305)	0.067
	rs182052	−0.284(−1.832–1.264)	0.719	1.341(-0.646–3.328)	0.186
DBP	rs16861194	0.531(−0.369–1.431)	0.247	−0.454(−2.495–1.587)	0.663
	rs266729	0.142(−0.676–0.960)	0.734	1.713(0.273–3.154)	0.020
	rs182052	0.47(−0.427–1.367)	0.305	1.122(−0.005–2.25)	0.051
FBG	rs16861194	−0.024(−0.230–0.182)	0.818	0.203(−0.461–0.868)	0.548
	rs266729	0.089(−0.104–0.281)	0.367	0.022(−0.336–0.381)	0.902
	rs182052	0.134(−0.080–0.347)	0.221	0.093(−0.149–0.335)	0.450
TG	rs16861194	0.019(−0.076–0.113)	0.697	0.004(−0.173–0.182)	0.963
	rs266729	0.014(−0.065–0.093)	0.730	0.090(−0.133–0.313)	0.430
	rs182052	0.117(0.015–0.218)	0.024	0.177(0.058–0.296)	0.003
TC	rs16861194	−0.001(−0.079–0.077)	0.975	0.045(−0.144–0.234)	0.641
	rs266729	−0.027(−0.099–0.045)	0.462	−0.068(−0.209–0.073)	0.344
	rs182052	0.092(0.012–0.171)	0.023	0.140(0.044–0.237)	0.004
HDL-C	rs16861194	0.004(−0.023–0.032)	0.755	0.025(−0.043–0.094)	0.468
	rs266729	−0.018(−0.043–0.007)	0.149	−0.02(−0.062–0.023)	0.370
	rs182052	0.009(−0.018–0.035)	0.534	0.040(0.007–0.073)	0.019
LDL-C	rs16861194	−0.006(−0.070–0.058)	0.863	0.039(−0.117–0.195)	0.623
	rs266729	−0.025(−0.082–0.033)	0.401	−0.053(−0.159–0.052)	0.322
	rs182052	0.064(0–0.128)	0.050	0.098(0.020–0.176)	0.014

**Table 3 genes-13-00161-t003:** The associations with lipid levels for the rs182052 ego and alter genotypes.

Trait	Ego Genotype	Alter Genotype	Frequency (%)	β(95%CI)	*P*
TG	AA	AA	564(15.6)	0 (reference)	
		AG	358(9.9)	−0.014(−0.199–0.170)	0.879
		GG	83(2.3)	−0.011(−0.243–0.22)	0.923
	AG	AA	358(9.9)	0.186(−0.002–0.375)	0.053
		AG	1036(28.6)	0.049(−0.088–0.186)	0.484
		GG	335(9.3)	0.061(−0.103–0.226)	0.463
	GG	AA	83(2.3)	0.248(−0.022–0.519)	0.072
		AG	335(9.3)	0.148(−0.029–0.325)	0.100
		GG	460(12.7)	0.084(−0.085–0.253)	0.328
TC	AA	AA	564(15.6)	0 (reference)	
		AG	358(9.9)	0.039(−0.121–0.199)	0.632
		GG	83(2.3)	0.001(−0.234–0.236)	0.995
	AG	AA	358(9.9)	0.133(−0.025–0.290)	0.098
		AG	1036(28.6)	0.080(−0.059–0.218)	0.259
		GG	335(9.3)	0.166(0.005–0.328)	0.043
	GG	AA	83(2.3)	0.146(−0.130–0.422)	0.299
		AG	335(9.3)	0.136(−0.026–0.297)	0.099
		GG	460(12.7)	0.212(0.035–0.388)	0.019
HDL-C	AA	AA	564(15.6)	0 (reference)	
		AG	358(9.9)	0.016(−0.038–0.069)	0.568
		GG	83(2.3)	−0.010(−0.090–0.070)	0.807
	AG	AA	358(9.9)	0.017(−0.035–0.070)	0.518
		AG	1036(28.6)	0.014(−0.035–0.064)	0.577
		GG	335(9.3)	0.049(−0.005–0.104)	0.076
	GG	AA	83(2.3)	−0.042(−0.120–0.037)	0.297
		AG	335(9.3)	0.048(−0.009–0.106)	0.097
		GG	460(12.7)	0.099(0.035–0.162)	0.002
LDL-C	AA	AA	564(15.6)	0 (reference)	
		AG	358(9.9)	−0.019(−0.144–0.106)	0.766
		GG	83(2.3)	−0.004(−0.184–0.176)	0.965
	AG	AA	358(9.9)	0.051(−0.076–0.178)	0.429
		AG	1036(28.6)	0.046(−0.065–0.156)	0.419
		GG	335(9.3)	0.118(−0.009–0.245)	0.069
	GG	AA	83(2.3)	0.064(−0.161–0.289)	0.579
		AG	335(9.3)	0.058(−0.071–0.188)	0.376
		GG	460(12.7)	0.177(0.037–0.317)	0.013

**Table 4 genes-13-00161-t004:** Effects of the alter rs182052 genotypes on ego lipid levels stratified by sex difference.

Trait	Genotypes	Same-Sex Siblings		Opposite-Sex Siblings	
		β(95%CI)	*P*	β(95%CI)	*P*
TG	AA/AG	0 (reference)		0 (reference)	
	GG	0.003(−0.105–0.112)	0.951	−0.090(−0.239–0.059)	0.236
TC	AA/AG	0 (reference)		0 (reference)	
	GG	0.055(−0.044–0.154)	0.278	0.055(−0.044–0.154)	0.278
HDL-C	AA/AG	0 (reference)		0 (reference)	
	GG	0.028(−0.007–0.063)	0.114	0.067(0.020–0.114)	0.005
LDL-C	AA/AG	0 (reference)		0 (reference)	
	GG	0.066(−0.012–0.144)	0.099	0.131(0.017–0.244)	0.024

**Table 5 genes-13-00161-t005:** Effects of the alter rs182052 genotypes on ego lipid levels stratified by specific sex.

Trait	Genotypes	Male-Ego with Female-Alter		Female-Ego with Male-Alter	
		β(95%CI)	*P*	β(95%CI)	*P*
TG	AA/AG	0 (reference)		0 (reference)	
	GG	−0.230(−0.441–−0.019)	0.033	0.043(−0.177–0.263)	0.700
TC	AA/AG	0 (reference)		0 (reference)	
	GG	0.117(−0.109–0.342)	0.311	0.089(−0.120–0.298)	0.405
HDL-C	AA/AG	0 (reference)		0 (reference)	
	GG	0.091(0.014–0.168)	0.021	0.045(−0.027–0.116)	0.224
LDL-C	AA/AG	0 (reference)		0 (reference)	
	GG	0.135(−0.044–0.313)	0.139	0.127(−0.038–0.292)	0.131

## Data Availability

The data presented in this study are available on request from the corresponding author. The data are not publicly available due to the policy of the Ethics Committee of the Peking University Health Science Center.

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
