# Peer review of "Indirect Genetic Effects of ADIPOQ Variants on Lipid Levels in a Sibling Study of a Rural Chinese Population"

_genes, 2022, doi:10.3390/genes13010161_

Round 1

Reviewer 1 Report

In their manuscript, the authors analyzed the correlation of serum lipid levels in sibling pairs and investigated the possible indirect effect of ADIPOQ variants between siblings. The concept of the article looks interesting, first of all, the authors' attempt to connect psychological parameters (assignment of brothers and sisters to ego and alter, depending on the influence on each other) with the direct or indirect influence of ADIPQ variants on lipid levels.

Nevertheless, there are a number of comments on the manuscript.

  1. I would like to clarify the concept of ego and alter. From the article it can be understood that the ego is an individual who is influenced by the people around him, i.e. his behavior can be modified by criticism / praise from another person. That is, the ego genotype is the genotypes of people who are susceptible to other people's opinions. In turn, an alter is an individual who, with his value judgment, can influence the behavior of another person, and he himself is little influenced and is not inclined to modify his behavior under external influence. Thus, the alter-genotype: these are the genotypes of those people who influence other people with their judgment, but themselves are not subject to such influence. Or is it not so? In any case, in the article to which the authors refer, it is noted that the ego is the subject being examined, and the alter is another subject with which he interacts, without specifying the nature of this interaction.
  2. In my opinion, the authors unreasonably apply the concept of indirect genetic influence in their work. In earlier studies, using the concept of ego and alter, the influence of psychosocial factors on obesity was assessed, the role of social connections was emphasized. It is not clear how the authors excluded the influence of factors from the external environment in them paper and explain the revealed connections precisely by indirect genetic influence. In this case, such an assumption looks too bold.

Reviewer 2 Report

The work is of interest and well discussed. The limitations of the study have been considered.

There are some minor spelling errors scattered throughout the text. The sentence “ for those who is male…”, lines 218-219, is not clear.

Round 2

Reviewer 1 Report

I received answers from the authors to the questions asked of me, I have no more comments